# HTNV infection of CD8$^+$ T cells is associated with disease progression in HFRS patients

Rongrong Liu[1,7], Ruixue Ma[1,7], Ziyu Liu[1,7], Haifeng Hu[2], Jiayi Shu[3], Peizhen Hu[4], Junjun Kang[5], Yusi Zhang[5], Mingwei Han[5], Xiaoxiao Zhang[1], Yiting Zheng[5], Qikang Ying[1], Shiyuan Hou[1], Wenqiu Wang[5], Fang Wang[1], Ning Cheng[6], Yan Zhuang[2], Jianqi Lian[2], Xia Jin [3✉] & Xingan Wu [1✉]

Hantaan viruses (HTNVs) are zoonotic pathogens transmitted mainly by rodents and capable of infecting humans. Increasing knowledge of the human response to HTNV infection can guide the development of new preventative vaccines and therapeutic strategies. Here, we show that HTNV can infect CD8$^+$ T cells in vivo in patients diagnosed with hemorrhagic fever with renal syndrome (HFRS). Electron microscopy-mediated tracking of the life cycle and ultrastructure of HTNV-infected CD8$^+$ T cells in vitro showed an association between notable increases in cytoplasmic multivesicular bodies and virus production. Notably, based on a clinical cohort of 280 patients, we found that circulating HTNV-infected CD8$^+$ T cell numbers in blood were proportional to disease severity. These results demonstrate that viral infected CD8$^+$ T cells may be used as an adjunct marker for monitoring HFRS disease progression and that modulating T cell functions may be explored for new treatment strategies.

[1] Department of Microbiology, School of Basic Medicine, Fourth Military Medical University, Xi'an, China. [2] Department of Infective Diseases, Tangdu Hospital, Fourth Military Medical University, Xi'an, China. [3] Shanghai Public Health Clinical Center, Fudan University, Shanghai, China. [4] State Key Laboratory of Cancer Biology, Department of Pathology, Xijing Hospital and School of Basic Medicine, Fourth Military Medical University, Xi'an, China. [5] School of Basic Medicine, Fourth Military Medical University, Xi'an, China. [6] Department of Otolaryngology-Head and Neck Surgery, University of California, San Francisco, CA 94115, USA. [7] These authors contributed equally: Rongrong Liu, Ruixue Ma, Ziyu Liu. ✉email: jinxia@shphc.org.cn; wuxingan@fmmu.edu.cn

Many emerging human infectious diseases are caused by zoonotic viruses[1,2], such as the deadly Ebola virus, highly pathogenic avian influenza viruses, severe acute respiratory syndrome (SARS) coronavirus (CoV), SARS-CoV-2[3,4], and Middle East respiratory syndrome CoV[5], as well as the subject of this study, hantavirus[6].

Hantaviruses belong to the *Bunyavirales* family of viruses that are mainly carried by rodents[7]. Some hantavirus species, such as the Hantaan virus (HTNV) prevalent in Asia, Dobrava virus, and Puumala virus (PUUV) prevalent in Europe, and Seoul virus worldwide, can cause hemorrhagic fever with renal syndrome (HFRS) in humans. Other hantavirus species, such as the Sin Nombre virus (SNV) and Andes virus (ANDV), that are prevalent in North and South America, respectively, can cause human hantavirus cardiopulmonary syndrome (HCPS)[8,9]. In addition, ANDV is known to be transmitted from person to person through the respiratory tract and saliva[10]. Infection with these viruses may cause various clinical presentations, from asymptomatic to acute fever and hemorrhage, acute kidney or lung injury, and even hypotensive shock[11]. HFRS cases in China account for more than 90% of the total global human infections, and they are caused by infection with HTNV, the prototype virus of the hantavirus family[6].

Host immune responses determine the outcome of human HTNV infection. Hantavirus has an incubation period of 2–4 weeks after infection, and by the time symptoms appear, an effective cellular immune response has usually been generated, producing specific antibodies[12]. Detection of hantavirus-specific IgM antibodies is usually used to confirm or exclude hantavirus infection[13]. Furthermore, a neutralizing antibody response is associated with favorable disease outcomes, i.e., high neutralizing antibody titers correlate with increased survival in hantavirus infection[14]. The activation and differentiation of T cells into effector cells that can produce cytokines is the main feature of adaptive immune response to various infections[15]. CD8+ T cell mediated immunity plays a critical role in clearing viruses by eliminating virus-infected cells. Virus-specific CD4+ and CD8+ T cells with polyfunctional cytokine responses and cytotoxic mechanisms are associated with the control of HTNV infection. The frequency of regulatory T cells in patients with HFRS is negatively correlated with the severity of the disease, which indicates that the inefficient control of T cell effector function may be the cause of more serious diseases[16]. However, too strong or too weak a T cell response has been linked to severe disease in both HCPS and HFRS[7]. Previous studies suggested that the increased capillary permeability might be caused by the attack of hantavirus-specific cytotoxic T cells (CTLs) on endothelial cells presenting viral antigens. Previous studies have shown that in HFRS, T cell response is negatively correlated with the severity of the disease, and the number of regulatory T cells that are considered to inhibit T cell response is negatively correlated with the severity of the disease. In rat experiments, hantavirus caused a persistent infection, and depletion of regulatory T cells helped to clear the virus in infected rats without causing immunopathology. These seemingly contradictory findings suggest a delicate balance between the T cell response of protection and immunopathogenesis. Therefore, elucidating the role of T cells in these diseases is important for better treatment and protection[16].

Patients with less severe HFRS usually show a greater degree of activation and proliferation of HTNV-specific CD4+ and CD8+ T cells, whereas patients with critical or severe HFRS often have limited T cell responses[17–19]. In addition, HFRS patients at the acute stage exhibit robust CTL responses against HTNV[18,20]. Moreover, a novel CD8+ T cell subset, namely, CD8low CD100− T cells, has been associated with virological control in patients with HFRS[21]. Previously, HTNV has been documented to infect endothelial cells[22], monocytes[23,24], and dendritic cells[25], and these infected cells can serve as antigen-presenting cells to activate CTLs with antiviral activities[25,26]. It remains unknown, however, whether HTNV directly interacts with human T cells during natural infection.

In this study, we started with demonstrating a considerable portion of CD8+ T cells in patients in the acute phase of severe HFRS contains the HTNV nucleocapsid protein (NP), and then confirmed that primary human CD8+ T cells can be infected with and support the full replication cycle of HTNV. We further showed that HTNV-infected primary human CD8+ T cell numbers are associated with disease stages. These results will be a useful guide to the development of biomarkers for monitoring HFRS disease progression, and that may provide insights into the design of new treatment strategies for HTNV.

## Results

**HTNV can infect CD8+ T cells in patients at the acute phase of HFRS.** A total of 280 blood samples was collected from 119 HFRS patients. The demographic information and clinical parameters collected during the hospitalization of all enrolled subjects are summarized in Table 1. According to the diagnostic criteria in the Prevention and Treatment Strategy of HFRS, the acute phase includes febrile, hypotensive, and oliguric stages; the convalescent phase includes diuretic and convalescent stages (Fig. 1a)[17].

To profile the peripheral T cell population in patients with HTNV infection, 122 freshly isolated peripheral blood mononuclear cell (PBMC) samples from HFRS patients were examined by flow cytometry, uninfected (healthy) donors as negative controls (Supplementary Fig. 4); sequential PBMC samples from some patients were also included to gain insight into T cell variations during disease progression. Leukocyte populations were classified into CD4+ T cell (CD3+ CD4+) and CD8+ T cell (CD3+CD8+) subsets and HTNV-infected cells were identified by intracellular staining of the HTNV NP. Our results showed that HTNV can infect CD8+ T cells, especially CD8low T cell subset, as demonstrated by representative FACS plots of sequential PBMC samples (on Days 10, 13, 19, and 23) from HFRS patient No. 10 (Fig. 1b). Over time, the percentage of HTNV NP-positive CD3+ CD8+ T cells decreased dramatically from 36.4 to 5.98%, whereas the percentage of CD3+CD4+ T cells positive for HTNV NP staining remained constant, between 1 and 3% (Fig. 1b).

Consistent with the previous reports[7,16], the percentage of total CD8+ T cells in the PBMC samples obtained from acute-phase patients was higher than that in samples from convalescent-phase patients (P < 0.001, Fig. 1c), and the reverse was true for CD4+ T cells (P < 0.001, Fig. 1c). The result was a decreased or even inverted CD4:CD8 ratio during the acute phase of HFRS. To investigate whether HTNV infection of CD8+ T cells is associated with disease progression, we divided the 122 patients into acute [n = 62, age range (27.0, 54.5), male/female (44/18)] and convalescent [n = 60, age range (33.8, 50.0), male/female (44/16)] phases. By examining the clinical and laboratory features of patients with HFRS, we stratified the patients into five stages or two disease phases (acute and convalescent) and found that age or gender does not co-segregate with patients' clinical features (Table 1). As illustrated in Fig. 1d, which presents the proportion of HTNV-infected CD8+ T cells and CD4+ T cells for each sample, an average of 13.1% (range: 0.15–82.1%) of the total CD8+ T cells infected by HTNV were in the acute phase (red), which was higher than the 1.64 % (range: 0.05–15%) in the convalescent phase (P < 0.001). The median (interquartile range, IQR) were 4.5% (1%, 20.98%) in the acute phase and 0.55% (0.21%, 1.6%), in the convalescent phase (P < 0.001). As shown in

**Table 1 Clinical and laboratory features of patients with HFRS stratified by five stages or two disease phases (acute and convalescent).**

| | Acute phase | | | Convalescent phase | |
|---|---|---|---|---|---|
| | Febrile | Hypotensive | Oliguric | Diuretic | Convalescent |
| | ($n = 45$) | ($n = 24$) | ($n = 75$) | ($n = 108$) | ($n = 28$) |
| Age (years) | 48.0 [27.8, 55.0] | 41.0 [34.5, 55.0] | 45.0 [29.0, 54.5] | 46.0 [29.0, 54.5] | 49.0 [29.8, 56.0] |
| Sex | | | | | |
| Male | 28 (62.2%) | 17 (70.8%) | 56 (74.7%) | 80 (74.1%) | 18 (64.3%) |
| Female | 17 (37.8%) | 7 (29.2%) | 19 (25.3%) | 28 (25.9%) | 10 (35.7%) |
| White blood cell count, ×10⁹/L | 12.6 [8.1, 16.3] | 14.1 [8.9, 33.0] | 10.9 [7.9, 16.0] | 8.1 [6.4, 10.1] | 7.1 [6.2, 9.0] |
| Platelet count, ×10⁹/L | 65.0 [40.8, 87.5] | 43.0 [25.0, 85.2] | 84.0 [53.0, 137.0] | 196.5 [107.5, 254.8] | 205.0 [144.5, 282.0] |
| Hemoglobin, g/L | 122.0 [111.8, 133.5] | 123.0 [93.5, 134.2] | 114.0 [98.0, 125.0] | 111.5 [100.0, 129.0] | 122.0 [103.5, 134.0] |
| Lymphocyte count, ×10⁹/L | 3.6 [2.4, 5.1] | 2.5 [1.8, 9.1] | 2.7 [1.5, 4.4] | 1.8 [1.3, 2.3] | 1.7 [1.4, 2.4] |
| Neutrophil count, ×10⁹/L | 6.7 [4.4, 8.8] | 5.5 [4.2, 12.8] | 7.3 [4.4, 9.4] | 4.8 [3.7, 6.1] | 4.1 [3.4, 5.2] |
| Alanine transaminase (ALT), U/L | 53.0 [37.5, 75.0] | 68.5 [46.8, 105.8] | 61.0 [44.0, 92.0] | 55.5 [34.0, 81.5] | 48.0 [36.5, 73.2] |
| Aspartate transaminase (AST), U/L | 55.0 [46.0, 76.0] | 90.5 [48.2, 164.8] | 57.0 [42.0, 120.0] | 43.0 [31.0, 61.0] | 33.5 [27.2, 49.5] |
| AST/ALT | 1.1 [0.9, 1.5] | 1.5 [1.0, 1.9] | 1.0 [0.8, 1.4] | 0.8 [0.6, 1.0] | 0.7 [0.6, 1.0] |
| Creatinine, µmol/L | 96.0 [72.1, 196.2] | 146.9 [105.2, 254.8] | 260.2 [167.1, 356.5] | 186.3 [104.2, 301.9] | 98.4 [66.2, 202.6] |
| Albumin, g/L | 28.4 [27.1, 31.2] | 31.6 [27.2, 33.7] | 31.1 [29.5, 34.4] | 36.0 [33.2, 40.7] | 36.2 [33.4, 38.7] |
| Uric acid, µmol/L | 302.8 [216.6, 446.0] | 251.6 [141.9, 350.9] | 217.9 [121.9, 415.6] | 431.9 [296.0, 533.8] | 390.0 [274.4, 454.4] |
| Blood urea nitrogen, mmol/L | 9.0 [5.2, 13.4] | 9.5 [7.1, 15.6] | 12.1 [8.6, 16.3] | 13.0 [7.7, 17.9] | 6.2 [5.1, 14.6] |
| Glomerular filtration rate, mL/min | 66.7 [35.2, 109.9] | 38.4 [20.8, 67.0] | 25.6 [15.8, 35.4] | 36.3 [19.2, 68.4] | 73.6 [22.8, 97.5] |
| Prothrombin time, s | 10.2 [9.9, 10.7] | 11.1 [10.6, 12.2] | 10.6 [10.1, 11.8] | 11.4 [10.7, 12.1] | 10.6 [10.5, 12.1] |
| Potassium, mmol/L | 3.4 [3.2, 3.8] | 3.7 [3.4, 3.9] | 3.6 [3.3, 4.0] | 3.6 [3.1, 3.9] | 3.8 [3.2, 4.1] |
| Sodium, mmol/L | 137.6 [133.1, 141.8] | 135.1 [132.3, 138.2] | 135.5 [133.6, 139.3] | 139.6 [136.6, 143.4] | 141.0 [138.7, 142.9] |

According to clinical practice, the acute phase should cover febrile, hypotensive, and oliguric stages and the convalescent phase diuretic and convalescent stages. Data are presented as the median (interquartile range, IQR) or n (%).
HFRS hemorrhagic fever with renal syndrome.

Supplementary Fig. 2, the patient samples were counted according to the stages of the disease; febrile, hypotensive, oliguric, diuretic, and convalescent. As the disease progressed, the number of the HTNV⁺CD8⁺ cells gradually decreased. In contrast, the proportion of HTNV-infected CD4⁺ T cells between acute and convalescent phases has no statistical differences ($P = 0.159$). Taken together, these data indicate that HTNV specifically infects CD8⁺ T cells. To provide more supporting information, we then sorted NP⁺ and NP⁻CD8⁺ T cells prepared from freshly isolated PBMC from patient H20-2, and measured viral mRNA expression in the two populations using qRT-PCR and found that the viral load of NP-positive CD8⁺ T cells was much higher than that of NP-negative cells (Supplementary Fig. 3).

**HTNV-infected CD8⁺ T cells produce infectious virions.** To confirm this observation of HTNV infection of CD8⁺ T cells, we next experimentally infected primary CD8⁺ T cells purified from PBMCs of healthy donors with HTNV in vitro at an MOI of 0.1 for 24, 48, 72, and 96 h and then performed intracellular staining of the HTNV NP to detect infected cells by immuno-fluorescence (IF) microscopy. At 48 h after infection, we detected HTNV-NP positive staining in the cytoplasm of CD8⁺ T cells; the staining was distributed homogeneously throughout the cells, except for a few patches of granular-shaped staining on the inner side of the plasma membrane (Fig. 2a). Notably, the same pattern of HTNV-NP IF staining was observed in peripheral CD8⁺ T cells from HFRS patients, as depicted by a representative result for patient No. H35-2 (Fig. 2b) and No.H24-2 (Supplementary Fig. 4).

To examine whether viral infection affects CD8⁺ T cell function, we quantified the amount of HTNV mRNA in CD8⁺ T cells by qRT-PCR. Consistent with the kinetics of viral protein expression, the amount of viral mRNA increased gradually from 0, 24, and 48 h post infection (h.p.i.), reached a peak at 72 h, and then decreased (Fig. 2c). As viral infection may trigger

activation of CD8⁺ T cells and apoptosis[27], variations in the levels of granzyme A (GZMA), granzyme B (GZMB), and perforin over the same period in the culture supernatant were detected by ELISA (Fig. 2c).

To further assess whether this viral infection is productive, we collected culture supernatants of infected CD8⁺ T cells at 0, 48, 72, and 96 h.p.i., added them to Vero E6 cells that are susceptible to HTNV infection, and then stained the cells for the HTNV NP 7 days later. The results confirmed an increasing trend of NP expression, indicating that infectious virions accumulated over time in the culture supernatants (Fig. 2d). Collectively, the above in vitro and ex vivo data demonstrate that HTNV can complete its replication cycle in CD8⁺ T cells.

**Production of infectious virions in CD8⁺ T cells involves viral particle transport by microvesicles.** To visualize HTNV replication more directly in CD8⁺ T cells, we used transmission electron microscopy (TEM) to examine key steps of the viral life cycle in association with ultrastructural changes in experimentally infected CD8⁺ T cells. Primary CD8⁺ T cells from healthy donors typically have a large nucleus and limited cytoplasm, a few organelle structures, and mitochondria, and a lack of microvesicles; their nuclei are located at the center and have a circular or elliptical shape and evenly distributed chromatin (Fig. 3aI). After HTNV infection, CD8⁺ T cells exhibited a notable increase in multivesicular bodies (MVBs) (Fig. 3aII) or microvesicles (Fig. 3aIII) in the cytoplasm and the emergence of exosome-like liposomal structures (Fig. 3aIV).

TEM also enabled visual tracking of the life cycle of HTNV replication. Viral entry occurred first by virion attachment to the cell surface (Fig. 3bI), followed by fusion between the viral envelope and cell membrane and the entry of nucleocapsids. After infection, virus particles were found in small aggregates in the cytoplasm with microvesicles (Fig. 3bII). For the majority of cells, no virus particles were detected on the cell surface; instead, the

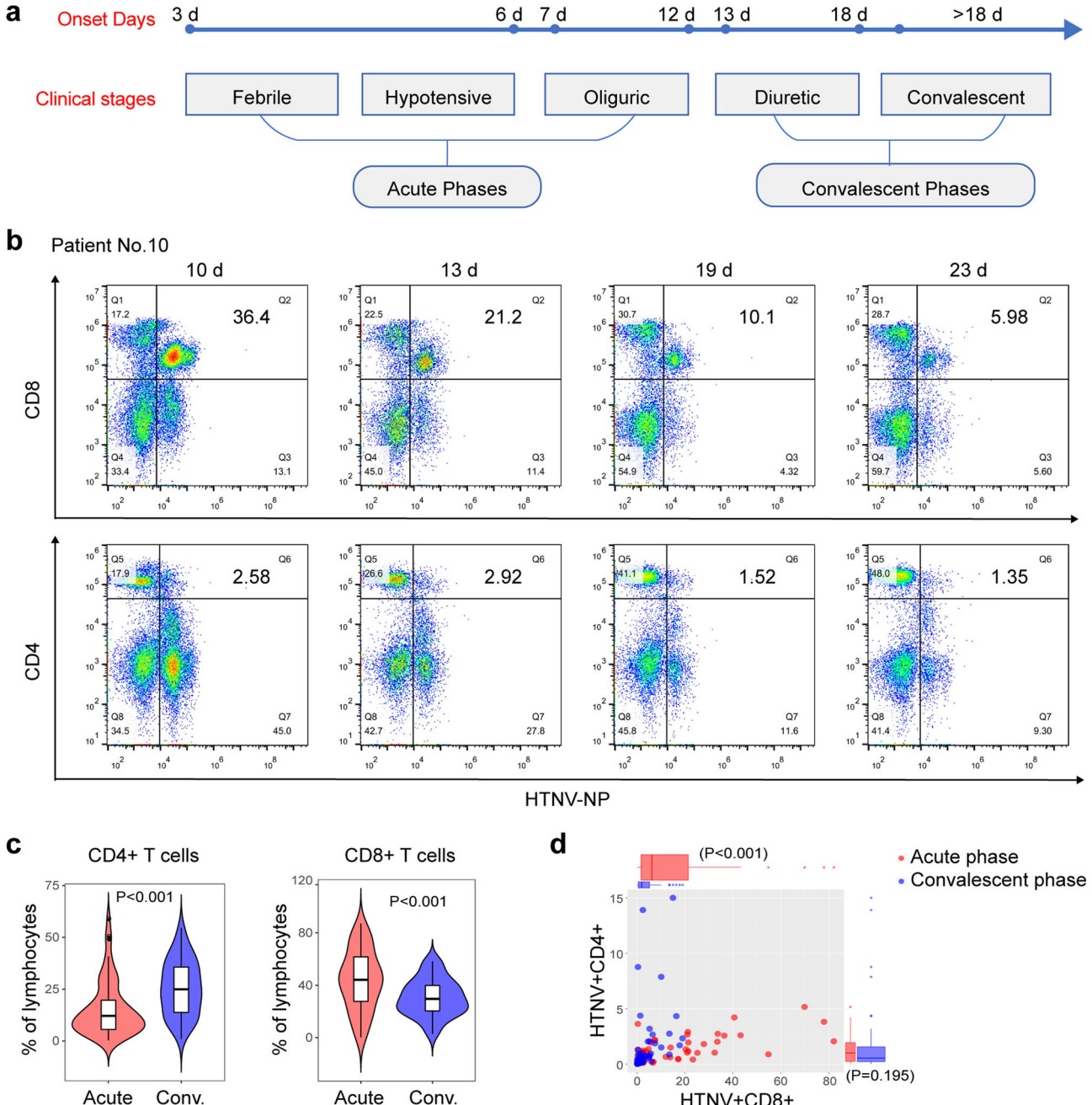

**Fig. 1 Characterization of virally infected T lymphocytes from HRFS patients at various stages of the disease. a** Classification of disease stages of HFRS patients. **b** Representative FACS plots of viral infection (HTNV-NP positive) of CD4+ and CD8+ T cells in PBMC samples obtained from HFRS patient No. 10 on days 10, 13, 19, and 23 after symptom onset. **c** Comparison of the percentages of CD8+ T cells (top panel) and CD4+ T cells (bottom panel) between the acute and convalescent phases of HFRS. (Acute, $n = 62$; Conv., $n = 60$). **d** Comparison between the frequency of HTNV-infected CD8+ T cells and CD4+ T cells during the acute and convalescent phases of HFRS. Each point represents a single sample; red indicates the acute phase and blues the convalescent phase. Groups were compared using a two-tailed unpaired Student's t-test. (Acute, $n = 62$; Conv., $n = 60$).

cells displayed virus-like particles in intracytoplasmic vacuoles. Dense granules ranging from 80 to 210 nm in diameter were also present in the cytoplasm and processes of the cells (Fig. 3bIII, IV). Nucleocapsids assembled in the vesicles, and some ribosomes were attached to the vesicle membrane. The nucleocapsids budded into smooth vesicles and acquired viral envelopes (Fig. 3bIII). However, these nucleocapsids were not yet enclosed by an envelope, as shown by the light-colored structure in the core. As the nucleocapsids replicated, the ribosomes attached to the surface of the vesicles initially decreased and gradually disappeared completely (Fig. 3bIV). At 96 h after infection, CD8+

T cell necrosis was detected, along with an increase in mature viral particles in the cytoplasm (Fig. 3bV).

Consistent with the data shown in Fig. 3b, the observed HTNV replication cycle can be summarized in a schematic illustration (Fig. 3c) that includes several key steps of the viral life cycle: (1) virus entry; (2) increased MVBs; (3) and (4) virus assembly and maturation; and (5) virion release.

**The proportion of HTNV-infected CD8+ T cells correlates with disease parameters and cytotoxic cytokine levels.** Having firmly

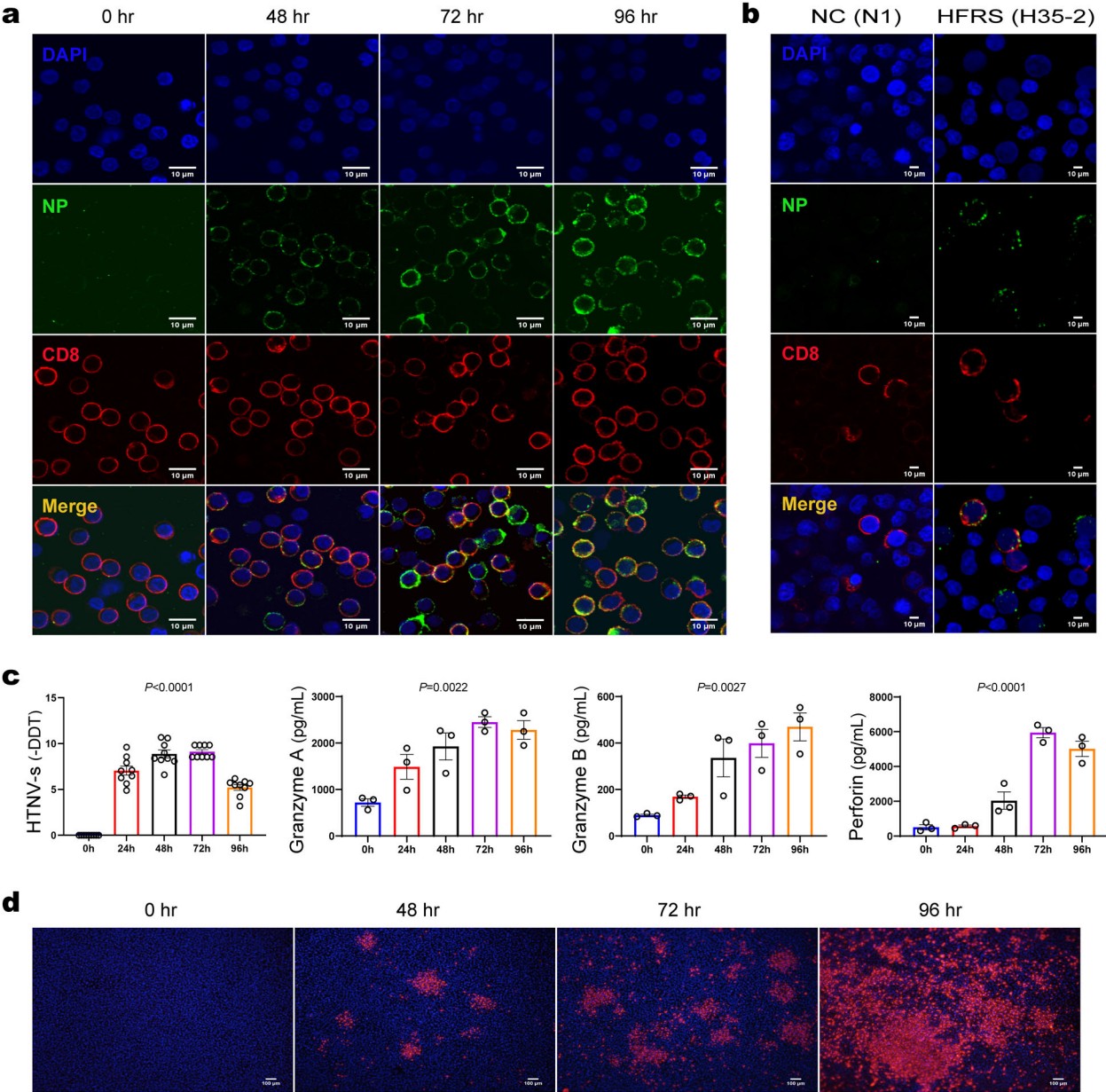

**Fig. 2 HTNV can infect primary human CD8+ T cells in vitro and promote their production of cytotoxic factors. a** Confocal microscopy identification of HTNV NPs in CD8+ T cells infected with HTNV. Positively selected CD8+ T cells from a healthy donor were infected with HTNV, collected at 0, 48, 72, and 96 h post infection, stained with either anti-HTNV-NP or anti-CD8a antibodies, and counterstained with DAPI (blue, top panel). Single-positive cells are either infected (HTNV NP, green, second panel) or CD8+ T cells (red, third panel); double-positive cells constitute an overlaid image of a single-positive cell and appear orange (bottom panel). (n = 3 cell cultures per experiment). **b** Identification of HTNV infection of CD8+ T cells (NP+CD8+) among PBMCs from an HFRS patient (H35-2). HTNV NP (green) or CD8 (red) single-positive cells, as well as NP+CD8+ double-positive (orange) cells, are shown. A normal control sample from an uninfected subject (N1) was included for comparison. **c** At 0, 24, 48, 72, and 96 h post infection, viral loads were measured by the detection of the HTNV s segment gene using quantitative real-time PCR, and β-actin was used as a housekeeping gene for normalization. (n = 9 cell cultures per experiment). The amounts of granzyme A, granzyme B, and perforin in culture supernatants were measured by ELISA. Values are expressed as the mean ± SEM from three independent experiments. One-way ANOVA with a post hoc Tukey test was used for multiple comparisons. (n = 3 cell cultures per experiment). **d** Culture supernatants from CD8+ T cells infected with HTNV were collected at 0, 48, 72, 96 h post infection and used to inoculate Vero E6 cells, which were stained for HTNV NP (red) after 7 days of culture. Scale bar = 10 μm. (n = 3 cell cultures per experiment).

established that HTNV can infect CD8+ T cells and that such infection alters the transcriptome of these cells, we investigated whether these observations are clinically relevant. White blood cell (WBC), lymphocyte (LYMPH), and neutrophil counts were elevated in acute-phase patients (from the febrile stage to the oliguric stage) and then declined to normal ranges as HFRS progressed to the convalescent phase (from the diuretic stage to the convalescent stage). Conversely, the platelet (PLT) count showed the opposite trend, and thrombocytopenia was detectable at the acute stage (Fig. 4a). CD8+ T cells related inflammatory cytokines including perforin, GZMA, and GZMB concentrations median (IQR) in the plasma of the corresponding patient samples is higher at the acute stage than at the convalescent stage (P < 0.001) (Fig. 4a). The levels of perforin and GZMB serum

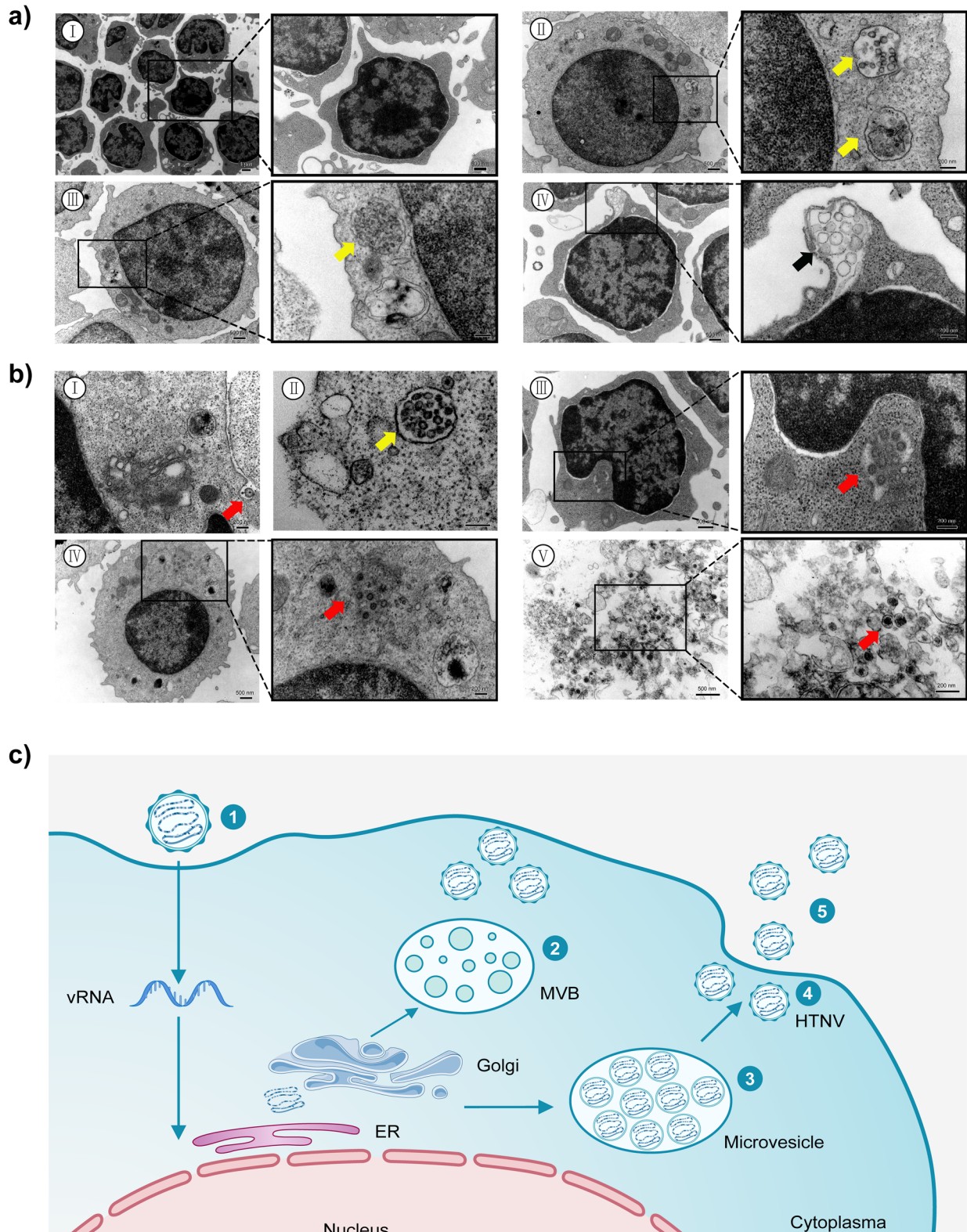

concentrations progressively increased from the febrile phase to the hypotensive phase and then gradually decreased during the convalescent phase (Supplementary Fig. 5).

To quantify the clinical significance of HTNV infection of CD8+ T cells, we assessed the correlation between CD8+ T cells and the levels of secreted cytokines and clinical parameters for all 122 patients. Results show that CD8+ T cell counts correlated positively with WBC ($r = 0.4$, $P < 0.01$) and LYMPN ($r = 0.6$, $P < 0.01$) but negatively with PLT ($r = -0.5$, $P < 0.01$). In addition, the CD8+ T cell ratio was correlated positively with HTNV-infected CD8+ T cell number ($r = 0.5$, $P = 0.0001$) and cytotoxic cytokine levels in the blood, including perforin ($r = 0.4$,

**Fig. 3 Ultrastructural identification of HTNV virions at different stages of the viral life cycle in primary human CD8+ T cells.** CD8+ T cells were infected with HTNV for 72 h and then fixed with 2% glutaraldehyde and analyzed by TEM. **a** Representative images of uninfected CD8+ T cells (I); representative images at high magnification revealed the presence of virus-like particles contained within multivesicular bodies (MBVs) (II and III, yellow arrows); the presence of vesicles secreted from the T cells (IV, black arrows). **b** Distribution of vesicle structures containing virion particles in the cytoplasm of HTNV-infected CD8+ T cells. Viral entry occurred first by virion attachment to the cell surface (I). Nucleocapsids are assembled in the vesicles, and some microvesicles formed in the vesicles (II). The nucleocapsids budded into the smooth vesicles and acquired viral envelopes (III). These nucleocapsids were not yet enclosed by an envelope, as shown by the light-colored structure in the core. As the nucleocapsids replicated, after which the ribosomes attached to the surface of the vesicles decreased initially and gradually disappeared completely (IV). At 96 h after infection, CD8+ T cell necrosis was detected, along with an increase in mature viral particles in the cytoplasm (V). The red arrows represent the virus. **c** Schematic illustration of the life cycle of hantaviruses in primary CD8+ T cells. Steps 1–5 correspond to the same stages in **b**.

$P = 0.0001$), GZMA ($r = 0.6$, $P < 0.01$), and GZMB ($r = 0.6$, $P < 0.01$) (Fig. 4b).

To determine predictive values for clinical parameters about prognosis, the receiver operating characteristic curve (ROC) and area under the curve (AUC) were analyzed. The PLT count had the highest diagnostic value (AUC = 0.91), followed by secreted GMZB (AUC = 0.84), perforin (AUC = 0.74), and LYMH (AUC = 0.71) levels. The ratio of CD8+ T cells (AUC = 0.72) to HTNV-infected CD8+ T cells showed a moderate diagnostic value (AUC = 0.70) (Fig. 4c). These results suggest that the proportion of HTNV-infected CD8+ T cells correlates with disease parameters and cytotoxic cytokine levels. Thus, CD8+ T cells can be used for immune function evaluation in HFRS patients.

## Discussion

Hantavirus mainly infects the capillary endothelial cells of the kidney and lung and induces a strong immune response in the human body[6]. However, why such immune responses fail to adequately control the disease remains uncertain. In this study, we discovered that human CD8+ T cells can be infected in vivo by HTNV and confirmed that primary human CD8+ T cells can be infected ex vivo to support the full replication cycle of HTNV. Moreover, completion of the HTNV replication cycle in CD8+ T cells involves virion transport by microvesicles. Remarkably, degrees of CD8+ T cell infection and dysfunction are associated with clinical markers of disease.

It is generally believed that HFRS and HCPS are caused by an uncontrolled systemic inflammatory response mediated by a variety of inflammatory cytokines, such as tumor necrosis factor-α, interleukin-8, and RANTES[6]. The inducer of such responses, however, is largely unknown. Previous studies have shown that hantavirus infection may induce a strong cellular immune response in humans, including an increase in the number of activated circulating CD8+T cells, infiltration of CD8+ T cells in the kidneys of patients with PUUV infection, and the persistence of virus-specific memory CD8+T cells even after HTNV or PUUV infection subsided[18]. The current study draws further attention to CD8+ T cells' contribution to pathogenesis.

Similar to patients with viral hemorrhagic fever[7], we observed that the number of CD8+ T cells increased substantially during HTNV acute infection and declined gradually to baseline after clearance of the virus. Previous studies have shown that CD8+ T cell responses specific for HTNV epitopes can be detected with pentamer staining, and a high proportion of CD8low T cells are present in the early stages of HTNV infection. Compared with the epitope-specific CD8high T cell subset, CD8low T cells secreted interferon-γ similarly, but proliferated less[18]. Moreover, some studies have shown that effector CD8+ T cells display the CD8low phenotype after virus infection or intracellular antigen stimulation[27–29]. In agreement with these reports, CD8lowCD100- T cells in HFRS patients express high levels of cytolytic effector

molecules and exhibit the phenotype of effector cells (CCR7+/−CD45RA−)[30].

Previous studies have described high frequencies of hantavirus-specific memory CD8+ T cells among PBMCs of individuals recovered from HTNV infection[31,32]. In a previous study in 1990[33], total blood LYMPHs were prepared from HFRS patients with by a density gradient on Ficoll-Hypaque. T and B cells were then purified by passing the total LYMPHs over a nylon wool column. Anti-HTNV monoclonal antibodies were used to detect viral antigen expression by IF analysis. T cells were infected with HTNV in the early stage of HFRS, and no specific fluorescence was observed in cells from the late diuretic stage to the convalescent phase.

The results suggest that virus replication in blood LYMPHs may partly contribute in the early stages to the impairment of cellular immune response and in vivo spread of HTNV to its target sites. Using this traditional method, the proportion of NP positive T cells in one patient was 52.4%, and that in the other patient was 65.2%[33]. These findings and the lack of evidence of human HTNV persistence or symptomatic reinfection, indicating the role of CD8 T cells in long-term protective immunity. Regardless, the differentiation process of virus-specific memory CD8+ T cells is far from clearly understood[17].

Our study has certain limitations. Although CD8low T cells are mostly HTNV-specific effector T cells, further classification of these cells may be achieved by using Viral-Track[34] and other phenotypic and genotypic markers. The regulatory role of each transcription factor has been investigated previously in other viral infections, and thus, our future research should focus on the identification of specific transcription factors in HTNV-infected human CD8+ T LYMPHs. Due to limitations in PBMC samples, we were unable to examine the secretion of specific cytokines by different T cell subsets.

In conclusion, we found in this study that CD8+ T cells were infected by HTNV in patients with acute HFRS. Moreover, a viral infection of CD8+ T cells caused their dysfunction in association with disease progression. These results will guide the development of biomarkers for evaluating immune functions in HFRS patients and suggest potential therapeutic targets for the treatment of HTNV.

## Methods

**Study design**. This study included 280 blood samples from 119 HFRS patients hospitalized in the Department of Infectious Diseases at the Tangdu Hospital of the Air Force Military Medical University (Xi'an, China) from October 2017 to January 2020. Two study cohorts were established, no.1 included 158 samples from 74 patients from November 2017 to January 2018, and no. 2 included 122 samples from 45 patients from November 2019 to January 2020.

HTNV infection was diagnosed by serological testing according to diagnostic criteria. To control for potential confounders, we excluded patients with autoimmune diseases, viral hepatitis, hematological diseases, diabetes, cardiovascular diseases, and other kidney or liver diseases. For in vitro assays, fresh peripheral blood was obtained from 12 healthy donors to isolate PBMCs. Before blood collection, all HFRS patients and healthy donors were informed and signed written consent[35].

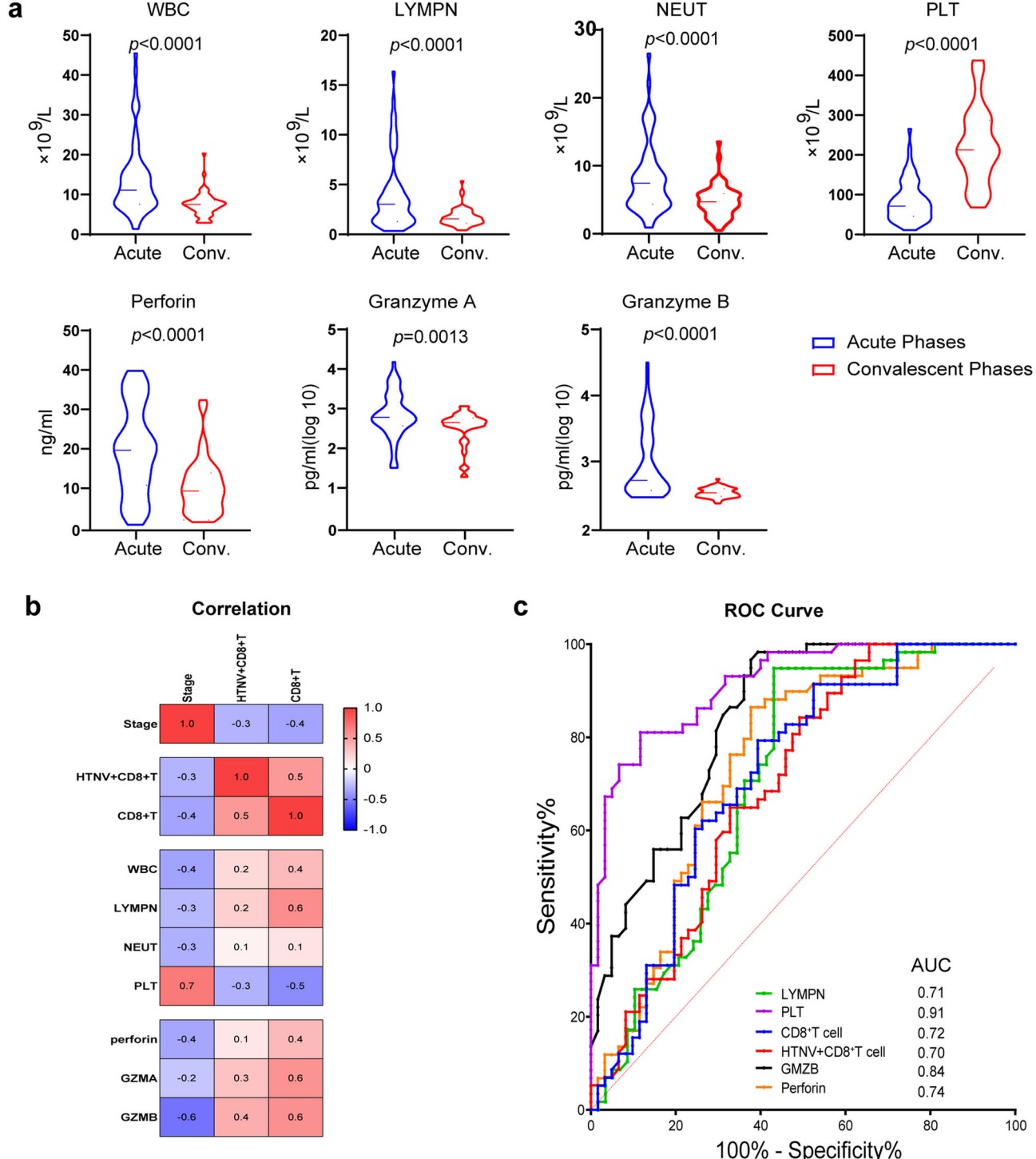

**Fig. 4 Correlations between HTNV-infected CD8+ T cells and clinical indices. a** Comparisons of white blood cell (WBC), lymphocyte (LYMPH), neutrophil (NEUT), and platelet (PLT) counts and levels of perforin, granzyme A (GZMA), and granzyme B (GZMB) between patients at the acute and convalescent phases of HFRS. Groups were compared using a two-tailed unpaired t-test. (Acute, $n = 62$; Conv., $n = 60$). **b** Correlations between CD8+ T cells or HTNV-infected CD8+ T cells and WBC, LYMPH, NEUT, PLT, perforin, GZMA, and GZMB. Correlations with $P < 0.05$ are indicated with an r value that ranged from −1 to 1. Negative correlations are shown in blue and positive correlations in red ($n = 122$). **c** Receiver operating characteristic curve (ROC) and area under the curve (AUC) analyses of the prognostic values of various clinical parameters for HFRS stages. The PLT count had the highest diagnostic value (AUC = 0.9149), followed by secreted GMZB (AUC = 0.8386) and perforin (AUC = 0.7448) levels, the WBC count (AUC = 0.7615) and LYMH count (AUC = 0.7111), and then the percentage of CD8+ T cells (AUC = 0.7195) or HTNV-infected CD8+ T cells (AUC = 0.7). (Acute, $n = 62$; Conv., $n = 60$).

**Flow cytometry**. To determine dynamic changes in T cell subsets throughout HFRS, the 122 PBMC samples from 45 HFRS patients were freshly isolated for flow cytometry analysis. According to the disease stage, the sample is divided into acute stage ($n = 62$) and convalescent stage ($n = 60$). All 122 PMBC samples were analyzed for CD3, CD4, CD8, and HTNV-NP expression by flow cytometry. Freshly isolated PBMCs were surface stained with monoclonal antibodies. CD8$^+$ T cells were quantified using APC/Cyanine7 anti-human CD3 Antibody (BioLegend. Cat. 300318), PE/Cyanine7 anti-human CD8 antibody (BioLegend. Cat. 344712); CD4$^+$ T cells were quantified using CD3-APC/Cy7 (BioLegend. Cat. 300318) and PE Cyanine5 anti-human CD4 Antibody (BioLegend. Cat.317412). The specific mAb 1A8 against HTNV-NP was used to identify HTNV-infected T cells; this mAb was prepared and purified by our laboratory[36]. For flow cytometry, mAb 1A8 was directly labeled with FITC (fluorescein isothiocyanate) following the protocol of a commercial kit (Lightning-Link® Antibody Labeling Kits, Innova Biosciences) developed for primary antibody labeling (TDR Biotech CO., LTD. Beijing China). HTNV-infected cells were identified by intracellular staining of the HTNV nucleocapsid protein (NP). Thoroughly resuspend cells and add 100 µL per well for microwell plates of Fixation/ Permeabilization solution for 20 min at 4 °C. Wash cells two times in 1× BD Perm/Wash™ buffer and pellet. (Cat. No. 554714). After 30 min of incubation in the dark with FITC-1A8 antibody, the cells were washed and finally re-suspended in 400 µL PBS and placed on FACS tubes for subsequent flow cytometry analysis. Mock-infected cells were treated equally and used for gating; 20,000 cells were counted from each condition.

Isotype control antibodies used in the original experiment were APC/Cy7 Mouse IgG2a, κ (BioLegend. Cat. 402206); PE/Cy7 Mouse IgG1, κ (BioLegend. Cat. 400126), PE/Cy5 Mouse IgG2b, κ (BioLegend. Cat. 402206), and FITC conjugated mouse IgG1, κ (BioLegend. Cat.400108). Flow cytometry compensation has been adjusted. There were ~$5 \times 10^5$ cells in each labeling reaction. The cells were incubated with antibodies at 4 °C for 40 min and detected using an ACEA NovoCyte™ flow cytometry (ACEA Biosciences Inc. China).

**Cell culture and viral infection**. Fresh peripheral blood from HFRS patients and healthy donors was used to isolate PBMCs, as reported previously. CD4$^+$ and CD8$^+$ T LYMPHs were rapidly purified from PBMCs by a magnetic cell sorting approach (Miltenyi Biotech Inc., Auburn, CA), strictly following the manufacturer's instructions (Miltenyi). The purified T LYMPHs were cultured in ImmunoCult™-XF T Cell Expansion Medium (STEMCELL, Vancouver, British Columbia, Canada). Vero E6 cells were purchased from American Type Culture Collection and cultured in DMEM supplemented with 10% fetal bovine serum. HTNV strain 76-118 was preserved in our laboratory. Vero E6 cells served as HTNV-sensitive cell lines and were used for virus propagation. Determination of virus infectivity titers was performed using a 50% tissue culture infective dose (TCID50) by immunofluorescence assays (IFAs).

For cell line infection, in pilot experiments, we examined various MOI titrations using the MTT assay and discovered that MOI = 0.1 is a suitable titer to achieve quantifiable infections, similar to published studies[36–38]. Specifically, we investigated HTNV 76-118 progeny production in primary T cells with three different MOI at various time points. For primary cell infection, purified CD8$^+$ T cells were plated at $5 \times 10^6$ cells/well in a six-well plate and then exposed to HTNV at MOI = 0.1, 0.5, 1 for 2 h, washed, and then harvested at different time points. The results were analyzed by qRT-PCR and shown as Fig. S3. We demonstrated that HTNV 76-118 production is dose-independent and mostly uniform within this MOI range, therefore chose the lowest virus input MOI of 0.1 in most experiments. (Supplementary Fig. 6)

Cells incubated with uninfected Vero E6 cell culture supernatants served as a negative control. After infection, purified T cells were collected at 0, 24, 48, 72, and 96 h.p.i. and used for quantification of viral RNA and morphological examinations. Culture supernatants of HTNV-infected T cells were collected at each time point and stored at −80 °C.

**Quantitative real-time PCR**. To confirm and quantify viral loads in HTNV-infected T cells, qRT-PCR was performed to detect the HTNV *s* segment gene expression level. Approximately $1 \times 10^7$ infected T cells were used for total RNA extraction, which strictly followed the manufacturer's instructions. Reverse transcription (RT) was performed to obtain complementary DNA (cDNA) by applying PrimeScript RT Master Mix (TaKaRa) following the manufacturer's instructions. Using the cDNA as a template, real-time quantitative PCR using a LightCycler 96 (Roche) was performed in the following steps: denaturation at 95 °C for 5 min and amplification for 45 cycles of 15 s at 98 °C, 30 s at 58 °C, and 30 s at 72 °C. Relative gene expression in terms of fold change was calculated using the $2^{-\Delta\Delta Ct}$ method.

**Enzyme-linked immunosorbent assay**. Sandwich ELISA (R&D Systems, USA) with two antibodies was carried out as described in the manufacturer's instructions to measure the concentration of cytotoxic mediators in HTNV-infected T cell culture supernatant. The following human ELISA kits were used: GZMA, GZMB, and Perforin. For the assay, aliquots of cell culture supernatant were added to the reaction mixture in 96-well strips from the kits, at 100 µL per well. Absorbance was measured at a wavelength of 450 nm using a Synergy HT ELISA plate reader (Biotec, Dresden, Germany). The concentration of cytotoxic proteins was determined based on the standard curve from each kit. All measurements were conducted in triplicate.

**Transmission electron microscopy**. To identify viral particles in HTNV-infected T cells, we used negative staining TEM to detect virus ultrastructure. HTNV-infected T cells were collected by centrifugation for the preparation of cell plumps, which were fixed in sodium cacodylate buffer containing 1% glutaraldehyde (0.2 M, pH 7.2) and postfixed in 1% osmium tetroxide. The fixed samples were then treated with an acetone solution for dehydration and embedded in epoxy resin. Finally, samples were polymerized at 60 °C for 3 days. The resin blocks were used for the preparation of ultrathin sections (50–70-nm thick). The sections were supported by copper grids, stained negatively by uranyl acetate and lead citrate (Electron Microscopy Science), and observed using a JEM100SX transmission electron microscope (JEOL, Tokyo, Japan).

**Immunofluorescence assay**. IFAs were performed to detect viral antigens in T cells. Cells were harvested, washed twice with PBS, and suspended at $1 \times 10^6$ cells/mL. The cells were mounted on gelatin-coated slides by liquid-based thin layer cytologic preparation; the slides were fixed with 4% paraformaldehyde for 25 min and permeabilized with 0.3% Triton-X100 for 30 min at room temperature. A blocking buffer containing 3% bovine serum albumin (Sigma) in PBS was used to block potential nonspecific binding sites at room temperature for 40 min. All slides were double-stained with anti-CD4 (Abcam) or CD8 (Abcam) antibodies and 1A8 to visualize HTNV-NP-positive CD4 or CD8 T cells, and fluorescence images were captured using the FLUOVIEW FV1000 confocal system (Olympus, Tokyo, Japan). Approximately 1000 cells from each group were examined, and cells displaying HTNV nucleocapsid protein were enumerated.

**Human subject studies**. This study was approved by the First Affiliated Hospital of Fourth Military Medical University IRB (KY20173166-1). Written informed consent was obtained from the patient before inclusion in the study.

**Statistics and reproducibility**. Data were analyzed for statistical significance using GraphPad Prism 8 software. Unless otherwise specified, all are represented as the mean ± the standard error of the mean (SEM). The data were analyzed by the Mann–Whitney $U$ test or Kruskal–Wallis test, as specified in the figure legends, using GraphPad Prism 9 software. Correlation analysis among detected traits is represented by Spearman rank correlation coefficients. A $P$ value of <0.05 was considered significant.

**Reporting summary**. Further information on research design is available in the Nature Research Reporting Summary linked to this article.

## Data availability
The authors declare that the data supporting the findings of this study are available with the paper and its supplementary information files. Source data for all figures are provided with the paper (Supplementary Data 1). Any remaining information can be obtained from the corresponding author upon reasonable request.

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

## Acknowledgements

We thank all the volunteers and collaborating clinicians for their participation.

## Author contributions

R.L. and X.M. designed the study. Z.H. and W.L. conceived the project. J.L., Y.L., Y.Z, and F.H. collected and processed the clinical specimens. M.H., Y.Z, Y.Q., W.W summarized the clinical data. Y.S., R.L., X.M.X.Z., and S.H., conducted the in vitro assays. J.K. was responsible for TEM. Y.Z. was responsible for flow cytometry. R.L. and X.M. drafted the manuscript. W.L., X.J., N.C., and X.W. drafted and revised the manuscript. All authors were involved in the discussion of the results and critical reading of the manuscript.

## Funding

This work was supported by grants from the National Natural Science Foundation of China (Nos. 81772167, 81971563, 81602494), the Key Research and Development Project of Shaanxi Province (No. 2019ZDLSF02-03), and the US NCI/NIH (K00 CA223019 to N.C.).

## Competing interests

The authors declare no competing interests.
