## [Peer Review File · Communications Biology]

Reviewers' comments:

Reviewer #1 (Remarks to the Author):

In the study described in this manuscript by Liu et al., they show that the CD8 primary T cells can be infected with hantaan virus (HTNV) and these cells support the full replication cycle of HTNV. Further they also did the transcriptomic analysis identify signature genetic changes in various subsets of infected cells. The manuscript is written fine and experiments mostly support the conclusions. However there are some major and minor points which can make the manuscript stronger upon addressing them.

Major comments

1. The most important point which is missing is assessing cytokine responses. Although authors show changes in CD8 T cells, functionality of the cells is crucial, mainly to deduce correlates. Authors do mention that the material was limited but from past experience and literature, I think cytokine response can be evaluated from the same samples. This is the main drawback of the manuscript which concerns me the most and needs to be verified experimentally.
2. Authors talk about cellular immune responses in context of HTNV. However there is no mention of the role of humoral immune responses against HTNVs. It should be mentioned in the introduction.
3. In the infection experiment, authors used MOI 0.1 for the experiments. What was the rationale of choosing this MOI. Did they do the MOI titration before?
4. Can authors comment on the correction of the responses with age or gender or the other parameters associated with the patients along with the patterns shown for the acute and convalescent phases? Was there a certain pattern observed?

Minor comments

1. Some references need to be added for the following statements,
 - Discussion line 278 "Hantaviruses primarily infect host capillary endothelial cells of various organs, especially those of the kidneys and lungs, and induce a robust immune response in 280 humans"
 - Discussion line 289 "It is widely accepted that HFRS and HCPS are caused by uncontrolled systemic inflammatory responses mediated by multiple inflammatory cytokines, including TNF- α , IL-8, and RANTES"
2. Introduction line 67 "however, too strong or too weak a T cell response has been linked to severe disease in both HCPS and HFRS". It should be "however, too strong or too weak of a T cell response has been linked to severe disease in both HCPS and HFRS".

Reviewer #2 (Remarks to the Author):

The manuscript by Liu et al demonstrates that HTNV that CD8+ T cells can be infected with HTNV. This finding is well supported by both ex vivo staining of CD8+ T cell from HFRS patients as well as in vitro infection of CD8+ T cells from healthy PBMCs, including TEM imaging of infected CD8+ T cells. The authors correlate the frequency of HTNV-infected CD8+ T cells with the acute and convalescent phase of disease as well as with clinical indices. The authors also use scRNAseq to characterise CD8+ T cell infection by HTNV. I believe this novel finding will be of interest to the readership of Communications Biology and would support its publication. However, the following issues must be addressed:

- 1) In figure 1, the authors need to show HTNV-NP staining from uninfected (healthy) donors to validate the specificity of the FACS staining. Staining with isotype control would also add validity to the method
- 2) Arrows should be added to the TEM images of figure 3 to help identify the morphological changes and cellular structures referred to in the text. Additionally, it would help if numbering/labelling was kept consistent: Fig 3A I is referred to as Fig3Aa in the text.

- 3) The scRNAseq experiments and their analysis are not adequately described or interpreted.
- It is not clear how infection status is accounted for in the analysis. Are all the CD8s that are sequenced presumed to be infected and can that be supported by any data? I would suggest the authors assess HTNV gene expression in each cell of their dataset to determine infection.
 - It is not clear how the differences in gene expression between CD8 subsets relate to HTNV infection. Relating to the comment above, were any infectrf CD8 cells analysed? Transcriptional differences between Tnaive, Tcm, Te and Tem are expected
 - Page 8 line 206 states PCA analysis was used however there are no PCA plots in the manuscript, only tSNE plots in fig 4.
 - The whole section on the RNAseq analysis needs to be revisited and clearly explain the samples used (what infected vs uninfected datasets are included) and the analysis needs to be clearly explained in relation to infected and uninfected cells (and then can be further divided in CD8 subsets).
- 4) The authors demonstrate that the frequency of HTNV-infected CD8 cells correlates with measures of clinical severity and with the acute phase of disease. Given the number of patients in each subgroup of acute (febrile, hypotensive, oliguric) and convalescent (diuretic, convalescent) samples, it would be of interest to determine how the frequency of HTNV-infected CD8 cells differs between those groups.

Minor comments:

Line 250 page 10: 'to quantity' should be corrected to 'to quantify'

Page 14 flow cytometry section: the HTNV-NP staining method needs further information, was this intracellular? What fix/perm methods have been used?

Reviewer #3 (Remarks to the Author):

This report by Liu et al. is a well-written investigation into immune responses and potential biomarkers of HTNV infection. As the authors nicely summarize, there is relatively little known regarding immunity and pathogenesis of HTNV infection, and thus, immunological insight as well as predictive biomarkers of disease hold widespread utility. The primary conclusions of this study are that HTNV infects CD8 T cells, and that HTNV-infected CD8 T cells are predictive of disease progression in HFRS patients. The authors should be commended for their relatively large dataset and an impressive amount of work, use of single-cell RNAseq, and some potentially novel and interesting findings. However, the overall robustness of the results suggesting CD8 T cells are infected with HTNV and that this is a relevant biomarker is limited.

1. Based on the evidence presented, the extent to which HTNV infects CD8 T cells in infected patients is unclear and not very convincing.

a. The crux of the authors' conclusion (that HTNV infects CD8 T cells and this is relevant for disease progression) here appears to be mainly derived from Fig. 1B. However, there are several issues of concern with this figure. The authors do not present proper controls. There should be staining from uninfected, healthy controls (in the representative plots in B as well as in the analysis in C to understand the level of background staining) as well as a negative staining control or FMO. It's hard to discern the level of non-specific binding occurring here, and 82.1% of all CD8 T cells in a patient being infected with HTNV seems extraordinarily high. Indeed, the frequency decreases over time in Fig. 1B which is encouraging (although there is no graphical/statistical demonstration of this and appears to be for just one individual), but so does the activation level and phenotype of CD8 T cells which can potentially change expression of nonspecific targets. Confidence in the staining would also be increased by sorting NP+ and NP- CD8 T cells and measure viral mRNA expression in the two populations through qPCR.

b. Related to the above point, Fig. 2B provides some confidence that this process is occurring in vivo and the authors have nicely included PBMCs from an uninfected control individual. However, in Fig 2B, the authors show two infected CD8 T cells, which appears to be 100% of the CD8 T (or at least 67%) cells in the field. There's no quantification of this for multiple individuals, so it's hard compare this with Fig. 1 and be fully confident that this is happening in vivo to the level the authors suggest.

c. Regarding the frequency of infected CD8 T cells: The authors report a range of 0.15-82% of HTNV-infected CD8 T cells in Fig. 1D, with a mean of 15.2%. In Fig. 2C-D, at 96h and ex vivo it appears 100% of all CD8 T cells are infected. Overall, it's hard to confidently grasp the level of CD8 T cell infection in vivo and in vitro and many of these frequencies seem very high (especially as the frequency of HTNV in vitro infected monocytes was reported to be ~1% in reference # 19 with nearly 100x MOI).

2. Related to the point above, the authors do a nice job in providing supporting data to their main conclusion in Figures 2&3. Figures 2/3 highlight that CD8 T cells can be infected by HTNV, and support production of infectious virions. These are interesting findings and, for the most part, well done experiments in these figures. However, in vitro infection findings do not necessarily indicate this happens in humans in vivo, and thus strengthening the findings from Fig1 or 2B are essential. These in vitro studies would have also been strengthened by comparing to either a population of cells known to produce infectious virus (i.e. a validated cell target of HTNV) and/or by comparing with infection of CD4 T cells (as the summary sentence in line 138 is that "HTNV preferentially infects CD8 T cells").

3. The main conclusion that HTNV-infected CD8 T cells could be a useful biomarker is also not very convincing as Fig 5B indicates that CD8 T cells are an equally robust marker (which has already been shown to correlate with disease stage), and there are other immunological parameters that correlate better with disease stage.

4. The authors do a nice job with the scRNAseq analysis and this is certainly a technique which could reveal some really interesting biology. However, it appears uninfected cells are not included in the analysis. Therefore, it's hard to determine what the impact of viral infection was. If the authors could distinguish infected and uninfected cells via viral mRNAs in their analysis, and then compare those populations this would be a very interesting piece of data (however it appears that nearly all cells are infected in vitro from Fig. 2).

5. The overall study design is unclear. There are 280 blood samples from 119 patients (line 105 and abstract) but then 122 PBMC samples are used for Fig 1. It's not clear how many blood samples are from the same patient? Also, having blood samples from the same patient throughout the course of infection (as other studies have done) would dramatically enhance the strength of the results as well.

Reviewer #1

Major comments

“In the study described in this manuscript by Liu et al., they show that the CD8 primary T cells can be infected with hantaan virus (HTNV) and these cells support the full replication cycle of HTNV. Further, they also did the transcriptomic analysis to identify signature genetic changes in various subsets of infected cells. The manuscript is written fine and experiments mostly support the conclusions. However, there are some major and minor points which can make the manuscript stronger upon addressing them.”

1. “The most important point which is missing is assessing cytokine responses. Although authors show changes in CD8 T cells, the functionality of the cells is crucial, mainly to deduce correlates. The authors do mention that the material was limited but from past experience and literature, I think cytokine response can be evaluated from the same samples. This is the main drawback of the manuscript which concerns me the most and needs to be verified experimentally.”

Answer:

Thank you for your suggestion. Cytokines are known to be important factors in HFRS. It is generally accepted that the pathogenesis of hantavirus infections is the result of virus-mediated host immune response. Among immune parameters, certain cytokines such as IL-1, IL-6, IL-10, and TNF- α were suggested to be involved in the pathogenesis, since increased levels of these cytokines were found in patients with HFRS.¹ The cytokines detected include TNF- α , IL-2, IL-6, and IFN- γ , which are produced by T cells and may mediate capillary leakage. Exposure to high doses of TNF- α in vivo is known to induce shock, capillary leakage, and mortality, and therapy with high doses of IL-2 causes an increase in vascular permeability¹.

As showed in Fig.5A, we have detected CD8⁺ T cells related to inflammatory cytokines including perforin, granzyme A, and granzyme B. The concentrations

median (IQR) in the plasma of the corresponding patient samples is significantly higher at the acute stage than at the convalescent stage ($P < 0.001$).

On this basis, we made further statistics according to the disease stage. In brief, the levels of perforin, and Granzyme B serum concentrations progressively increased from the febrile phase to the hypotensive phase and then gradually decreased during the convalescent phase. These results had been added as Figure.S2 in the revised manuscript.

The description has been added in the results section.

(page 10, line 265-271)

“ CD8+ T cells related inflammatory cytokines including perforin, granzyme A, and granzyme B concentrations median (IQR) in the plasma of the corresponding patient samples is significantly higher at the acute stage than at the convalescent stage

($P < 0.001$) (Figure 5A). The levels of perforin and Granzyme B serum concentrations progressively increased from the febrile phase to the hypotensive phase and then gradually decreased during the convalescent phase (Figure.S2)”.

2. “Authors talk about cellular immune responses in the context of HTNV. However, there is no mention of the role of humoral immune responses against HTNVs. It should be mentioned in the introduction.”

Answer:

A paragraph on the cellular immune responses in HTNV has been added to the introduction in the revised text.

(page 3-4, line to 65-93)

“Host immune responses determine the outcome of human HTNV infection. Hantavirus infections have an incubation period of approximately 2–4 weeks after transmission and an effective immune response has usually already developed at the onset of symptoms, including high levels of specific antibodies.¹² Measurement of hantavirus-specific IgM antibodies was usually used for the diagnosis or exclusion of hantavirus infection.¹³ Furthermore, a neutralizing antibody response is associated with disease outcomes. Clinical research has shown that high neutralizing antibody titers correlate with increased survival in hantavirus infection.¹⁴ T cell activation and differentiation into cytokine-producing effector cells are a major feature in adaptive immunity to various infectious agents.¹⁵ T-cell mediated immunity plays a critical role in clearing viruses by eliminating virus-infected cells. Virus-specific CD4⁺ and CD8⁺ T-cells with polyfunctional cytokine responses and cytotoxic mechanisms were associated with HTNV infection control. The frequencies of regulatory T cells in HFRS patients are inversely correlated with disease severity, suggesting that inefficient control of T cell effector functions may be responsible for more severe disease.¹⁶ However, too strong or too weak of a T cell response has been linked to severe disease in both HCPS and HFRS⁷. We previously hypothesized that increased capillary permeability observed in HFRS may be caused by hantavirus-specific cytotoxic T cells that attach endothelial cells presenting viral antigens on their

surface-based on clinical observations and *in vitro* experiments. In HFRS, in one report, contrary to HCPS, T cell responses negatively correlated with disease severity, but in another report, the number of regulatory T cells, which are thought to suppress T cell responses, negatively correlated with disease severity. In rat experiments, in which hantavirus causes persistent infection, depletion of regulatory T cells helped infected rats clear virus without inducing immunopathology. These seemingly contradictory findings may suggest delicate balance in T cell responses between protection and immunopathogenesis. Both too strong and too weak T cell responses may lead to severe disease. It is important to clarify the role of T cells in these diseases for better treatment and protection.¹⁶”

3. “In the infection experiment, authors used MOI 0.1 for the experiments. What was the rationale of choosing this MOI. Did they do the MOI titration before?”

Answer:

For infection, in pilot experiments, we examined various MOI titrations using the MTT assay and discovered that MOI=0.1 is a suitable titer to achieve quantifiable infections, similar to published studies³⁻⁵.

Specifically, we investigated HTNV 76-118 progeny production in primary T cells with three different MOI at various time points. For infection, purified CD8⁺ T cells were plated at 5×10^6 cells/well in a 6-well plate and then exposed to HTNV at MOI=0.1,0.5,1 for 2 h, washed, and then harvested at different time points. The results were analyzed by qPCR and shown as **Figure.S3** in the revised manuscript. We demonstrated that HTNV 76-118 production is dose-independent and mostly uniform within this MOI range, therefore chose the lower virus input MOI of 0.1 in most experiments.

4. “Can authors comment on the correction of the responses with age or gender or the other parameters associated with the patients along with the patterns shown for the acute and convalescent phases? Was there a certain pattern observed?”

Answer:

We have analyzed our data and summarized as descriptive statistics in the form of **table S1**, by examining the clinical and laboratory features of patients with HFRS stratified by five stages or two disease phases (acute and convalescent), and found age or gender do not co-segregate with patients’ clinical features.

Table S1. Age and gender showed no correlation with the clinical feature in the study patient cohort

	Acute phase			Convalescent phase	
	Febrile	Hypotensive	Oliguric	Polyuric	Convalescent
	(n=45)	(n=24)	(n=75)	(n=108)	(n=28)
Age (years)	48.0 [27.8, 55.0]	41.0 [34.5, 55.0]	45.0 [29.0, 54.5]	46.0 [29.0, 54.5]	49.0 [29.8, 56.0]
Sex					
Male	28(62.2%)	17(70.8%)	56(74.7%)	80(74.1%)	18(64.3%)
Female	17(37.8%)	7 (29.2%)	19(25.3%)	28(25.9%)	10(35.7%)

Minor comments:

1. Some references need to be added for the following statements,

- Discussion line 278 “Hantaviruses primarily infect host capillary endothelial cells of various organs, especially those of the kidneys and lungs, and induce a robust immune response in 280 humans”

- Discussion line 289 “It is widely accepted that HFRS and HCPS are caused by uncontrolled systemic inflammatory responses mediated by multiple inflammatory cytokines, including TNF- α , IL-8, and RANTES”

Answer:

A reference “Liu, R. et al. Vaccines and Therapeutics Against Hantaviruses. Front. Microbiol. 10, (2020)” has been inserted following the above-mentioned text. (Page 10, Line 293 and Page 10, Line 303).

3. Introduction line 67 “however, too strong or too weak a T cell response has been linked to severe disease in both HCPS and HFRS”. It should be “however, too strong or too weak of a T cell response has been linked to severe disease in both HCPS and HFRS”.

Answer:

This has been modified in the text accordingly. (Page 4, Line 80)

Reviewer #2

Major comments:

The manuscript by Liu et al demonstrates that HTNV that CD8⁺ T cells can be infected with HTNV. This finding is well supported by both ex vivo staining of CD8⁺ T cells from HFRS patients as well as in vitro infection of CD8⁺ T cells from healthy PBMCs, including TEM imaging of infected CD8⁺ T cells. The authors correlate the frequency of HTNV-infected CD8⁺ T cells with the acute and convalescent phase of disease as well as with clinical indices. The authors also use scRNAseq to characterize CD8⁺ T cell infection by HTNV. I believe this novel finding will be of interest to the readership of Communications Biology and would support its publication. However, the following issues must be addressed:

1. “In figure 1, the authors need to show HTNV-NP staining from uninfected (healthy) donors to validate the specificity of the FACS staining. Staining with isotype control would also add validity to the method.”

Answer:

Isotype control antibodies were used in the original experiment. A more detailed description was added to the revised methods to make this clear.

(Page 13. Line 379-383).

“ CD8⁺ T cells were quantified using APC/Cyanine7 anti-human CD3 Antibody (BioLegend. Cat. 300318), PE/Cyanine7 anti-human CD8 Antibody (BioLegend. Cat. 344712); CD4⁺ T cells were quantified using CD3-APC/Cy7 (BioLegend. Cat. 300318) and PE Cyanine5 anti-human CD4 Antibody (BioLegend. Cat.317412). ”

(Page 14. Line 390-393).

“ Isotype control antibodies used in the original experiment were APC/Cy7 Mouse IgG2a, κ (BioLegend. Cat. 402206); PE/Cy7 Mouse IgG1, κ (BioLegend. Cat. 400126), PE/Cy5 Mouse IgG2b, κ (BioLegend. Cat. 402206), and FITC conjugated mouse IgG1, κ (BioLegend. Cat.400108).”

We have used the same method to test the HTNV-NP staining from uninfected (healthy) donors as negative controls as suggested, an experiment of which is shown as **Figure.S4** in the revised submission (Page 5. Line 123-124).

2. Arrows should be added to the TEM images of figure 3 to help identify the morphological changes and cellular structures referred to in the text. Additionally, it

would help if numbering/labeling was kept consistent: Fig 3A I is referred to as Fig3Aa in the text.

Answer:

Several arrows have been added to the electron microscope picture (Figure 3A and 3B). The yellow arrows represent multivesicular bodies (MVBs) (Figure 3Ab)(Figure 3Bb) or microvesicles (Figure 3Ac), the black arrows represent exosome-like liposomal structures (Figure 3Ad), and the red arrows represent the virus (Figure 3B a,c,d,e). The image tags have been modified in results (page 7, line 184-202) and figure legend (page 24, line 657-674).

3) The scRNAseq experiments and their analysis are not adequately described or interpreted.

a. It is not clear how infection status is accounted for in the analysis. Are all the CD8s that are sequenced presumed to be infected and can that be supported by any data? I would suggest the authors assess HTNV gene expression in each cell of their dataset to determine infection.

Answer:

To gain insight into the potential functional alterations occurring in HTNV-infected

CD8⁺ T cells, we performed scRNA-seq using CD8⁺ T cells sorted by positive selection with magnetic beads from PBMCs. We then infected the cells with HTNV (MOI=0.1) for 48 h. Not all of the CD8⁺ T cells were infected, but a part of them did. In PBMC of patients, it can be seen that the infected T cells are a cluster of CD8^{low} cells.

Because scRNA-seq technologies rely on polyadenylated RNA isolation and amplification, current scRNA-seq methods can, in theory, detect these viral RNA programs and therefore enable accurate identification of the infected cells and their unique properties at single-cell resolution. However, in the case of human clinical samples, these tools are limited, making the pathogen infected cells and viral reservoir cell types hard to detect. While such an approach has already been used to study both *in vitro*^{6,7} and *in vivo* infection models⁸, no general computational framework has been developed to detect viruses and analyze host-viral maps in clinical samples. In part, this is due to the absence of a 5' cap structure and a 3' poly(A) tail at the termini of the negative-strand RNA genomes, common features usually generated by RNA polymerase II transcription. It has been particularly difficult to develop reverse genetics systems for bunyaviruses⁹. However, its impact is dependent on several factors, the most critical one being the biochemical and pathophysiological properties of the virus. The absence of a poly(A) tail at the end of viral RNA molecules can significantly decrease their capture rate efficiency in current scRNA-seq techniques.

However, HTNV does not have a poly A tail, so it cannot be detected in the original 10X scRNA-seq data. This may hinder the ability to robustly identify infected cells or discern differential expression between infected and bystander cells in such viruses.

For this reason, we especially supplemented the experiment and designed an HTNV-specific probe, and scRNA-seq libraries were constructed according to Singleron GEXSCOPE® protocol (Singleron Biotechnologies). This method is still under development. Another limiting factor is the potential scarcity of viral reads and infected cells in the sample. As shown in our analysis of HTNV-infected samples (Figure below), only a limited number of viral reads were detected in some of the samples. In the preliminary experiment, this probe captured the virus in some of the T cells, which has certain suggestive significance.

“Supplementary Methods

Hantaan Virus Probe design

According to the Hantaan Virus sequence of NCBI, the probe sequence was designed for the conserved regions of Hantaan Virus M and S genes respectively. The length of the probe sequence is 25-30bp. The probe sequence is coupled with the capture magnetic beads containing the molecular tag. In this way, the same bead contains polyT probe and Hantaan Virus capture probe, so that the host transcriptome and virus transcriptome information can be captured at the same time, and the cell barcodes of the two types of probes are also consistent, which can accurately locate the virus-expressing cell.

Single-cell RNA sequencing

Single-cell suspensions with 1×10^5 cells/mL in concentration in PBS (HyClone) were prepared. Single-cell suspensions were then loaded onto microfluidic devices and scRNA-seq libraries were constructed according to Singleron GEXSCOPE® protocol by GEXSCOPE® Single-Cell RNA Library Kit (Singleron Biotechnologies)⁴¹. Individual libraries were diluted to 4nM and pooled for sequencing. Pools were sequenced on Illumina HiSeq X with 150 bp paired-end reads.

scRNA-seq quantifications and statistical analysis

Raw reads were processed to generate gene expression profiles using an internal pipeline. Briefly, after filtering read one without poly T tails, cell barcode and UMI

was extracted. Adapters and poly A tails were trimmed (fastp V1) before aligning read two to GRCh38 with ensemble version 92 gene annotation (fastp 2.5.3a and featureCounts 1.6.2)⁴². Reads with the same cell barcode, UMI, and gene were grouped together to calculate the number of UMIs per gene per cell. The UMI count tables of each cellular barcode were used for further analysis. Cell type identification and clustering analysis using Seurat program^{43,44}. The Seurat program (<http://satijalab.org/seurat/>, R package, v.3.0.1) was applied for the analysis of RNA-Sequencing data. UMI count tables were loaded into R using read.table function. Then we set the parameter resolution to 0.6 for the FindClusters function to clustering analyses. Differentially expressed genes (DEGs) between different samples or consecutive clusters were identified with function FindMarkers. GO function enrichment analysis was performed on the gene set using the cluster profile software to find biological functions or pathways that are significantly associated with the genes specifically expressed⁴⁵.”

b. It is not clear how the differences in gene expression between CD8 subsets relate to HTNV infection. Relating to the comment above, was any infection of CD8 cells analyzed? Transcriptional differences between Tnaive, Tcm, Te, and Tem are expected

Answer:

Current single-cell sequencing techniques (Singleron Biotechnologies) can only capture a fraction of cells with the virus, therefore, unable to directly analyze the transcriptome differences between infected and uninfected cells.

c. Page 8 line 206 states PCA analysis was used however there are no PCA plots in the manuscript, only tSNE plots in fig 4.

Answer:

PCA was performed using variable genes, which were selected based on average expression and dispersion. Clusters and t-SNE plots are presented based on selected PCA dimensions. This content has been modified in the revised text.

page 8, line 221-223 in the results section.

“ tSNE plots was used to reduce the dimensions of similarity analysis among the cells and resulted in their partitioning into 14 clusters (Figure 4A) ”

page 17, line 482-484 in the methods section.

“ PCA was performed using variable genes, which were selected based on average expression and dispersion. Clusters and t-SNE plots are presented based on selected PCA dimensions.”

d. The whole section on the RNAseq analysis needs to be revisited and clearly explain the samples used (what infected vs uninfected datasets are included) and the analysis needs to be clearly explained in relation to infected and uninfected cells (and then can be further divided into CD8 subsets).

Answer:

This part has been carefully revised with the addition of a new section on a new single-cell detection method with a virus probe. However, current single-cell sequencing techniques (Singleron Biotechnologies) can only capture a fraction of cells with the virus, therefore, unable to directly analyze the transcriptome differences between infected and uninfected cells.

4) The authors demonstrate that the frequency of HTNV-infected CD8 cells correlates with measures of clinical severity and with the acute phase of the disease. Given the number of patients in each subgroup of acute (febrile, hypotensive, oliguric) and convalescent (diuretic, convalescent) samples, it would be of interest to determine how the frequency of HTNV-infected CD8 cells differs between those groups.

Answer:

“ As shown in **Figure.S5**, the patient samples were counted according to the stages of the disease; febrile, hypotensive, oliguric, diuretic, and convalescent. As the disease progressed, the number of the HTNV⁺CD8⁺ cells gradually decreased.”

The above paragraph has been inserted into the revised results section. (see **Page 6**, **line 147-150**)

“ In a previous study in 1990¹⁰, total blood lymphocytes were prepared from HFRS patients with Ficoll- Hypaque gradient. T and B cells were then purified by passing the total lymphocytes over a nylon wool column. Detection of viral antigen by immunofluorescence assay using monoclonal antibodies to HTNV showed the T cells were infected by HTNV during the early stages of HFRS and no specific fluorescence was seen in the cells from the late diuretic phase to the convalescent phase. The results suggest that virus replication in blood lymphocytes may partly contribute in the early stages to the impairment of cellular immune response and in vivo spread of HTNV to its target sites. Using this traditional method, we quantified the proportion of NP positive T cells in one patient as 52.4%, and that in the other patient as 65.2%¹⁰. This phenotype is consistent with our conclusion. ”

The above paragraph has been inserted into the revised discussion section (page 11-12, line 324-334)

Minor comments:

“ Line 250 page 10: ‘to quantity’ should be corrected to ‘to quantify’

Page 14 flow cytometry section: the HTNV-NP staining method needs further information, was this intracellular? What fix/perm methods have been used?”

Answer:

According to reference 10, The specific experimental steps have been inserted into the original text .

(Page 14, Line 386-394)

“ HTNV-infected cells were identified by intracellular staining of the HTNV nuclear protein (NP). The specific steps of this procedure have been added to the material and method section (Page 13, line 373-381). In brief, thoroughly resuspend cells and add 100 μ L per well for microwell plates of Fixation/ Permeabilization solution for 20 minutes at 4°C. Wash cells twice with 1 \times BD Perm/Wash™ buffer and pellet (Cat. No. 554714). After 30 min of incubation in the dark with FITC-1A8 antibody, the cells were washed and finally re-suspended in 400 μ L PBS and placed on FACS tubes for subsequent flow cytometry analysis. Mock-infected cells were treated similarly and used for gating; 20,000 cells were counted from each condition.”

Reviewer #3

Major comments:

This report by Liu et al. is a well-written investigation into immune responses and potential biomarkers of HTNV infection. As the authors nicely summarize, there is relatively little known regarding immunity and pathogenesis of HTNV infection, and thus, immunological insight, as well as predictive biomarkers of disease, hold widespread utility. The primary conclusions of this study are that HTNV infects CD8 T cells and that HTNV-infected CD8 T cells are predictive of disease progression in HFRS patients. The authors should be commended for their relatively large dataset and an impressive amount of work, use of single-cell RNAseq, and some potentially novel and interesting findings. However, the overall robustness of the results suggesting CD8 T cells are infected with HTNV and that this is a relevant biomarker is limited.

1. “Based on the evidence presented, the extent to which HTNV infects CD8 T cells in infected patients is unclear and not very convincing.

1) .The crux of the authors’ conclusion (that HTNV infects CD8 T cells and this is relevant for disease progression) here appears to be mainly derived from Fig. 1B.

However, there are several issues of concern with this figure. The authors do not present proper controls. There should be staining from uninfected, healthy controls (in the representative plots in B as well as in the analysis in C to understand the level of background staining) as well as a negative staining control or FMO. It's hard to discern the level of non-specific binding occurring here.”

Answer:

Isotype control antibodies were used in the original experiment. A more detailed description was added to the revised methods to make this clear.

(Page 14. Line 390-393).

“ Isotype control antibodies used in the original experiment were APC/Cy7 Mouse IgG2a, κ (BioLegend. Cat. 402206); PE/Cy7 Mouse IgG1, κ (BioLegend. Cat. 400126), PE/Cy5 Mouse IgG2b, κ (BioLegend. Cat. 402206), and FITC conjugated mouse IgG1, κ (BioLegend. Cat.400108).”

We have used the same method to test the HTNV-NP staining from uninfected (healthy) donors as negative controls as suggested, an experiment of which is shown as **Figure.S4** in the revised submission (Page 5. Line 123-124).

2) *“82.1% of all CD8 T cells in a patient being infected with HTNV seems extraordinarily high. Indeed, the frequency decreases over time in Fig. 1B which is encouraging (although there is no graphical/statistical demonstration of this and appears to be for just one individual), but so does the activation level and phenotype of CD8 T cells which can potentially change the expression of nonspecific targets.”*

Answer:

As we described in the results (page 6 , line 148-156), Figure 1D presents a representative of the proportion of HTNV-infected CD8+ T cells and CD4+ T cells in a particular sample. On average, 13.1% (range: 0.15% to 82.1%) of the total CD8+ T cells infected by HTNV were in the acute phase (red), which was still significantly higher than the 1.64 % (range: 0.05% to 15%) in the convalescent phase (P<0.001). The median (IQR) were 4.5% (1%,20.98%) in the acute phase and 0.55% (0.21%,1.6%), in the convalescent phase (P<0.001). The difference was indeed considerable.

Figure 1D is the statistical result of all samples' flow data.

3) *“Confidence in the staining would also be increased by sorting NP+ and NP- CD8 T cells and measure viral mRNA expression in the two populations through qPCR.”*

Thank you for your suggestion. We tried this method on freshly isolated patient H20-2 this year and found a viral load of NP-positive CD8 T cells was higher than that of NP-negative cells (Figure below). We will add this method to future research. The samples from previous studies last year have all been used for flow cytometry and cannot be sorted.

b. *“Related to the above point, Fig. 2B provides some confidence that this process is occurring in vivo and the authors have nicely included PBMCs from an uninfected control individual. However, in Fig 2B, the authors show two infected CD8 T cells,*

which appear to be 100% of the CD8 T (or at least 67%) cells in the field. There's no quantification of this for multiple individuals, so it's hard to compare this with Fig. 1 and be fully confident that this is happening in vivo to the level the authors suggest."

Answer:

Fig. 2B is a representative graph. We have tested another group of patient's PBMC (H24-2) and found the staining was the same (Figure S6). The degree of infection differs in patients with HFRS. Immunofluorescence results only play a representative role and are not convenient for statistics. However, flow cytometry is more representative.

c. Regarding the frequency of infected CD8 T cells: The authors report a range of 0.15-82% of HTNV-infected CD8 T cells in Fig. 1D, with a mean of 15.2%. In Fig. 2C-D, at 96h and ex vivo, it appears 100% of all CD8 T cells are infected. Overall,

it's hard to confidently grasp the level of CD8 T cell infection in vivo and in vitro and many of these frequencies seem very high (especially as the frequency of HTNV in vitro infected monocytes was reported to be ~1% in reference # 19 with nearly 100x MOI).

Answer:

In Figure 2D, we collected culture supernatants of infected primary CD8⁺ T cells at 0, 48, 72, and 96 h post-infection, added them to Vero E6 cells that are susceptible to HTNV infection, and then stained the cells for the HTNV-NP protein 7 days later. The results indicating that infectious virions accumulated over time in the culture supernatants of CD8⁺ T cells. Combined with Figure 2C, the data demonstrate that HTNV can complete its replication cycle in CD8⁺ T cells. In HFRS patients, T cells are not only primitive, so they cannot be compared.

Very high infection levels were also observed in a previous study in 1990¹⁰, in which total blood lymphocytes were prepared from HFRS patients by a density gradient on Ficoll- Hypaque. T and B cells were then purified by passing the total lymphocytes over a nylon wool column. Detection of viral antigen by immunofluorescence assay using monoclonal antibodies to HTNV showed the T cells were infected by HTNV during the early stages of HFRS and no specific fluorescence was seen in the cells from the late diuretic phase to the convalescent phase. The results suggest that virus replication in blood lymphocytes may partly contribute in the early stages to the impairment of cellular immune response and in vivo spread of HTNV to its target sites. Using this traditional method, the proportion of NP positive T cells in one patient was 52.4%, and that in the other patient was 65.2%¹⁰. This phenotype is consistent with our conclusion.

The MOI in reference # 19 is 7.5 for PUUV which is very different from our HTNV. There are many MOI identification methods, and the virus quantification method of our research group has been used in many articles^{4,5}. For infection, we did the MOI titration and MTT before, because of the small size of T cells, a virus of 10⁷ TCID₅₀/ml, the amount of virus needed by the 5×10⁶ cells/well is 71.4ul. Based on reference³⁻⁵, MOI=0.1 is a suitable titer.

In this experimental study, we investigated HTNV 76-118 progeny production in primary T cells with three different MOI at determined time points. For infection, purified CD4⁺ and CD8⁺ T cells were plated at 5×10⁶ cells/well in a 6-well plate and then exposed to HTNV at MOI=0.1,0.5,1 for 2 h. The results were analyzed by qPCR (Figure.S2). Use of lower MOI (0.1) could result in accurate, precise quantitative assays in virus diagnosis and titration methods².

2. Related to the point above, the authors do a nice job in providing supporting data to their main conclusion in Figures 2&3. Figures 2/3 highlight that CD8 T cells can be infected by HTNV, and support the production of infectious virions. These are interesting findings and, for the most part, well-done experiments in these figures. However, in vitro infection findings do not necessarily indicate this happens in humans in vivo, and thus strengthening the findings from Fig1 or 2B are essential. These in vitro studies would have also been strengthened by comparing to either a population of cells known to produce infectious virus (i.e. a validated cell target of HTNV) and/or by comparing with infection of CD4 T cells (as the summary sentence in line 138 is that “HTNV preferentially infects CD8 T cells”).

Answer:

These are valid suggestions. After comparing HTNV susceptible cells HUVEC, THP-1, and A549, we found that the number of viruses entering the cell and their ability to replicate is different. We modified our wording and used “preferentially”

instead of “specifically” in the revised text. (Page 6. Line 152)

3. The main conclusion that HTNV-infected CD8 T cells could be a useful biomarker is also not very convincing as Fig 5B indicates that CD8 T cells are an equally robust marker (which has already been shown to correlate with disease stage), and there are other immunological parameters that correlate better with disease stage.

Answer:

The HTNV-infected CD8 T cells are a part of total CD8 T cells, Thrombocytopenia is a typical clinical manifestation of HFRS, while the increase of GMZB is closely related to CTL function, but the proportion of infected T cells can better reflect the relationship between the course of disease and prognosis. In viral infectious diseases, it is accepted that the proportion of CD8 T cells is related to the progression of the disease, but not all viruses can infect CD8 T cells directly, and it is related to disease course. This is also a new phenomenon that we have discovered. However, we appreciate the reviewer’s suggestion and therefore change the papers’ title from “predictive” of disease stages with strongly indicate the potential as a biomarker to a more modest “associated with” disease stages.

4. The authors do a nice job with the scRNAseq analysis and this is certainly a technique that could reveal some really interesting biology. However, it appears uninfected cells are not included in the analysis. Therefore, it’s hard to determine what the impact of viral infection was. If the authors could distinguish infected and uninfected cells via viral mRNAs in their analysis, and then compare those populations this would be a very interesting piece of data (however it appears that nearly all cells are infected in vitro from Fig. 2).

Answer:

To gain insight into the potential functional alterations occurring in HTNV-infected CD8⁺ T cells, we performed scRNA-seq using CD8⁺ T cells sorted by positive selection with magnetic beads from PBMCs. We then infected the cells with HTNV (MOI=0.1) for 48 h. Not all of the CD8⁺ T cells were infected, but a part of them did.

In PBMC of patients, it can be seen that the infected T cells are a cluster of CD8^{low} cells.

Because scRNA-seq technologies rely on polyadenylated RNA isolation and amplification, current scRNA-seq methods can, in theory, detect these viral RNA programs and therefore enable accurate identification of the infected cells and their unique properties at single-cell resolution. However, in the case of human clinical samples, these tools are limited, making the pathogen infected cells and viral reservoir cell types hard to detect. While such an approach has already been used to study both *in vitro*^{6,7} and *in vivo* infection models⁸, no general computational framework has been developed to detect viruses and analyze host-viral maps in clinical samples. In part, this is due to the absence of a 5' cap structure and a 3' poly(A) tail at the termini of the negative-strand RNA genomes, common features usually generated by RNA polymerase II transcription. It has been particularly difficult to develop reverse genetics systems for bunyaviruses⁹. However, its impact is dependent on several factors, the most critical one being the biochemical and pathophysiological properties of the virus. The absence of a poly(A) tail at the end of viral RNA molecules can significantly decrease their capture rate efficiency in current scRNA-seq techniques.

However, HTNV does not have a poly A tail, so it cannot be detected in the original 10X scRNA-seq data. This may hinder the ability to robustly identify infected cells or discern differential expression between infected and bystander cells in such viruses.

For this reason, we especially supplemented the experiment and designed an HTNV-specific probe, and scRNA-seq libraries were constructed according to Singleron GEXSCOPE® protocol (Singleron Biotechnologies). This method is still under development. Another limiting factor is the potential scarcity of viral reads and infected cells in the sample. As shown in our analysis of HTNV-infected samples (Figure below), only a limited number of viral reads were detected in some of the samples. In the preliminary experiment, this probe captured the virus in some of the T cells, which has certain suggestive significance.

In all, current single cell sequencing techniques (Singleron Biotechnologies) can only capture a fraction of cells with the virus, therefore, unable to directly analyze the transcriptome differences between infected and uninfected cells.

5. The overall study design is unclear. There are 280 blood samples from 119 patients (line 105 and abstract) but then 122 PBMC samples are used for Fig 1. It's not clear how many blood samples are from the same patient? Also, having blood samples from the same patient throughout the course of infection (as other studies have done) would dramatically enhance the strength of the results as well.

Answer:

We apologize for having not described this clearly before. In the revised paper, we have rephrased to accurately describe the study design.

(page 13,line 362-367)

“ This study included 280 blood samples from 119 HFRS patients. Two study cohorts were established, No.1 included 158 samples from 74 patients from November 2017 to January 2018, and No.2 included 122 samples from 45 patients from November 2019 to January 2020. “

(page 13,line 375-378)

“ To determine dynamic changes in T cell subsets throughout the course of HFRS, the

122 PBMC samples from 45 HFRS patients were freshly isolated for flow cytometry analysis. According to the disease stage, the sample is divided into acute stage (n=62) and convalescent stage (n=60). ”

Reference:

1. Saksida, A. Serum levels of inflammatory and regulatory cytokines in patients with hemorrhagic fever with renal syndrome. *8* (2011).
2. Abdoli, A. & Jamali, A. Determining Influenza Virus Shedding at Different Time. *CELL J.* **15**, 6 (2013).
3. Ma, H. *et al.* The Long Noncoding RNA NEAT1 Exerts Antihantaviral Effects by Acting as Positive Feedback for RIG-I Signaling. *J. Virol.* **91**, (2017).
4. Ma, H.-W. *et al.* In-Cell Western Assays to Evaluate Hantaan Virus Replication as a Novel Approach to Screen Antiviral Molecules and Detect Neutralizing Antibody Titers. *Front. Cell. Infect. Microbiol.* **7**, (2017).
5. Wang, K. *et al.* The Glycoprotein and Nucleocapsid Protein of Hantaviruses Manipulate Autophagy Flux to Restrain Host Innate Immune Responses. *Cell Rep.* **27**, 2075-2091.e5 (2019).
6. Shnayder, M. *et al.* Defining the Transcriptional Landscape during Cytomegalovirus Latency with Single-Cell RNA Sequencing. *mBio* **9**, (2018).
7. Drayman, N., Patel, P., Vistain, L. & Tay, S. HSV-1 single-cell analysis reveals the activation of anti-viral and developmental programs in distinct sub-populations. *eLife* **8**,
8. Steuerman, Y. *et al.* Dissection of Influenza Infection In Vivo by Single-Cell RNA Sequencing. *Cell Syst.* **6**, 679-691.e4 (2018).
9. Flick, K. *et al.* Rescue of Hantaan virus minigenomes. *Virology* **306**, 219–224 (2003).
10. Gu, X. S. *et al.* Hemorrhagic fever with renal syndrome. Separation of human peripheral blood T and B cells and detection of viral antigen. *Chin. Med. J. (Engl.)* **103**, 25–28 (1990).

Reviewers' comments:

Reviewer #1 (Remarks to the Author):

The review is satisfactory. No further comments.

Reviewer #2 (Remarks to the Author):

The authors have addressed my concerns

Reviewer #3 (Remarks to the Author):

The authors have addressed all of my concerns except for one major and one minor issue.

Major: While the authors should be commended on their efforts of identifying HTNV infected cells through scRNAseq, this hasn't changed the issue with Figure 4 lacking proper controls (uninfected cells). As it stands, the figure is comparing the transcriptome between very well characterized populations of CD8 T cells, and thus the data provides no new information on T cells or information related to HTNV. The authors mention in the rebuttal (and show in Fig 1) that HTNV infected T cells are CD8low. Perhaps they could use this as a proxy for discriminating infected and non-infected cells in the scRNAseq data? Currently, it appears the CD8 low cells aren't included in the analysis and are classified as "other" (Fig. 4 B,C). CD8a expression levels or another proxy for infected cells could be validated in the recently generated scRNAseq data (included in rebuttal). Then it might be possible to draw conclusions about CD8 T cells infected and not infected with HTNV, providing insight into how HTNV infection changes the CD8 T cell transcriptome.

Minor: Related to my initial concerns on the robustness of data demonstrating HTNV infects CD8 T cells, the authors have done a nice job of validating the results with inclusion of staining of healthy controls and qPCR of sorted HTNV infected cells. However, this qPCR data (included in the rebuttal) should be included in the manuscript as it validates their approach.

Reviewer #3

Major comment

1, *“While the authors should be commended on their efforts of identifying HTNV infected cells through scRNAseq, this hasn’t changed the issue with Figure 4 lacking proper controls (uninfected cells). As it stands, the figure is comparing the transcriptome between very well-characterized populations of CD8 T cells, and thus the data provides no new information on T cells or information related to HTNV. The authors mention in the rebuttal (and show in Fig 1) that HTNV infected T cells are CD8low. Perhaps they could use this as a proxy for discriminating infected and non-infected cells in the scRNAseq data. Currently, it appears the CD8 low cells aren’t included in the analysis and are classified as “other” (Fig. 4 B, C). CD8a expression levels or another proxy for infected cells could be validated in the recently generated scRNAseq data (included in rebuttal). Then it might be possible to draw conclusions about CD8 T cells infected and not infected with HTNV, providing insight into how HTNV infection changes the CD8 T cell transcriptome.”*

Response:

We thank the reviewer for these thoughtful suggestions, and agree that Figure 4 as it currently stands is not conclusive. We therefore take the suggestion to remove this figure from the revised paper. Follow up studies will be performed to examine this question more thoroughly.

As a matter of scientific interests, we would like to elaborate a bit on the rationale behind making such a decision. There are notable technical difficulties that led us decide not to pursuit this question immediately. Figure 1 shows the expression level of HTNV-NP protein as detected by flow cytometry in the form of high and low fluorescence intensity. Whereas the single cell sequencing technique quantifies high and low mRNA levels. To detect “low level” expression by either technique is difficult, draw comparison between the two low levels is not likely to yield convincing results.

Even though we tried a new protocol (Singleron GEXSCOPE® protocol) to

capture infected cells, only a small number of viral infected cells were captured, making it very hard to directly analyze the transcriptome differences between infected and uninfected cells.

As the reviewer correctly pointed out, even large single-cell sequencing companies such as 10x and Singleron are unable to solve this problem satisfactorily.

Based on above, we think not to include Figure 4 is a sensible thing to do at this moment.

Minor comment

“Related to my initial concerns on the robustness of data demonstrating HTNV infects CD8 T cells, the authors have done a nice job of validating the results with inclusion of staining of healthy controls and qPCR of sorted HTNV infected cells. However, this qPCR data (included in the rebuttal) should be included in the manuscript as it validates their approach.”

Response:

We thank the reviewer for this suggestion. We have added the referred results as Fig. S3 in the revised manuscript. These results clearly show that viral load of NP-positive CD8 T cells was much higher than that of the NP-negative cells. A more detailed description of this was added to the revised results section (Page 6, line 164-169), as the follows:

“ We then sorted NP⁺ and NP⁻CD8⁺ T cells prepared from freshly isolated PBMC from patient H20-2, and measured viral mRNA levels in these two cell populations using qPCR. Results showed that the viral load of NP-positive CD8 T cells was much higher than that of the NP-negative cells (Supplementary Figure 3). ”